# Pathways to Nursing and Midwifery Education in Tanzania with Reflection to the Global Perspectives: A Narrative Review

**DOI:** 10.3390/nursrep15120452

**Published:** 2025-12-18

**Authors:** Tumbwene Elieza Mwansisya, Mary Apolinary Lyimo, Eunice Siaity Pallangyo

**Affiliations:** School of Nursing and Midwifery, The Aga Khan University, Dar es Salaam P.O. Box 125, Tanzania; mary.lyimo@aku.edu (M.A.L.); eunice.siaity@aku.edu (E.S.P.)

**Keywords:** nursing pathways, nursing education, nursing and midwifery profession, nursing regulations, generation Z

## Abstract

**Background/Objectives**: This review aimed to explore the pathways of nursing and midwifery education in Tanzania and compare them with global perspectives. The goal was to identify similarities, differences, and areas for potential improvement to align with international standards. **Methods**: A narrative literature review was carried out through databases with published studies in nursing and gray literature. The database search included Medline, PubMed, Google Scholar, EBSCO, PsycINFO, clinical nursing, and gray literature from January 2014 up to December 2024. The search process was carried out by the authors with the following key words: admission, pathway to nursing profession, delivery mode, generative dynamic, and learning models. The search strategy included studies from selected countries in East Africa, Europe, North America, Australia, and Asia. The inclusion criteria were (1) published papers or reviews addressing the review topics; (2) studies published in the English language; (3) gray literature on the reviewed topics; (4) studies originating from Tanzania, East Africa, Europe, Asia, Australia, and North America. The selected countries served as a source for comparison of nursing and midwifery education in Tanzania with the globe. **Results**: A total of 758 articles were reviewed from the selected databases. Finally, 27 studies and 11 gray literature sources were included. In Tanzania, the overall duration of nursing education from primary education to a bachelor’s degree for diploma graduates is relatively long. Students complete approximately 14 years of schooling from primary education to the completion of a diploma, including three years at the diploma level. To enroll in a degree program, candidates are required to have two years of professional experience, followed by four years of academic training and a mandatory one-year internship. Globally, the duration of educational programs is generally decreasing due to generational shifts and advancements in technology. **Conclusions**: Whilst nursing and midwifery education is regulated in Tanzania, the current admission criteria and duration of the program do not align with the global standards. Future studies that provide the comparison of curriculums among universities in Tanzania with global standards would provide a deep understanding of the competencies, teaching models, learning environment, duration, and desired learning outcomes for nursing and midwifery education.

## 1. Background

Globally, nursing and midwifery services are estimated to comprise over 80% of the healthcare workers (HCW) worldwide [1] and play a significant role in the healthcare system [2]. These professions have evolved over the years to respond to technology, generation dynamics, changes in models of healthcare delivery, and a growing emphasis on evidence-based practice [3]. The transition of nurses and midwives to a higher level of registration has been linked to an increase in the skills-mix required in healthcare delivery [1]. Consistently, the Sustainable Development Goals (SDG 3 target 3) stress the development of HCW with adequate training, education, and skills that are relevant to the needs of the population [4]. While the majority of developed countries resorted to innovative educational programs that develop competent nurses and midwives [5,6,7], many lower-income countries (LICs) have a mismatch between education strategies, health system requirements, and community demands [1]. Therefore, the utilization of research evidence in recruitment, admission, and production of the nursing and midwifery profession is of paramount importance to strengthen the healthcare system.

The World Health Organization [8] in its report on global standards stated that the future goal for an initial education of professional nurses and midwives should be a university degree. In some high-income countries, the baccalaureate degree has been established as the minimum entry to the nursing and midwifery profession. For example, Belgium, the United Kingdom, New Zealand, Canada, Australia, Spain, Italy, and Norway have recently made the baccalaureate degree a requirement for nursing and midwifery registration [9]. The reform of nursing education to university education has also been observed in some African countries [10]. The baccalaureate degree has been found to attract high-quality students, reduce medical profession domination, and allow the autonomous nursing profession [10]. Studies indicate that, given a positive learning environment for nursing and midwifery students, no matter the mode of entry or their school grades, students can achieve good academic performance [11]. Contrarily, in Tanzania, the entry criteria to the nursing and midwifery profession are stringent and hinder the new entrants to the nursing and midwifery profession.

Studies in Tanzania have widely reported the acute shortages of healthcare workers in health facilities [12,13] and dissatisfaction with care related to nurse-midwives’ low competencies, poor communication, and lack of respect and dignity for clients and families [14]. In East Africa, Tanzania included, the production of competent nurses and midwives has been limited by a lack of program options and access issues [15]. Around the globe, a secondary school education with specific subjects, such as biology and chemistry, is required for direct entry into the nursing and midwifery profession, and for a university degree, a diploma is required [9,16,17]. However, in Tanzania, enrollment to the available nursing and midwifery programs, among others, has been limited by prohibitive entry criteria (Physics, Chemistry, Biology, English, and Mathematics as mandatory subjects) and long duration of study (4 years to both direct entry and those with Diploma) accompanied by one-year mandatory internship even to the practicing registered nurses and midwives. These requirements limit the career development of interested nurses and midwives [18]. Therefore, choosing the most appropriate entry point is likely to increase enrollment for nursing and midwifery students and give hope to registered nurses and midwives for their career development.

Typically, nursing and midwifery students acquire theoretical education in schools and practice in hospitals and communities. Recently, there has been a dramatic change in health information technology, patients and disease profiles, demand for healthcare systems, and a change in nursing roles [19,20,21] requiring innovative education delivery models to cope with changes [21]. High-level innovative education programs have been found to promote positive healthcare provider behaviors such as problem-solving, performance of complex functions, and effective communication [22,23]. For example, since 2002, the Aga Khan School of Nursing and Midwifery (SONAM) has developed a work-study program by using the foundation of global evidence on the competency of nurses and midwives with careful consideration of the contextual demands and human resources shortages [24]. The program is conducted in such a way that students gain theoretical knowledge for two days a week and use this knowledge immediately thereafter at clinical placement, preferably where they are employed. Competent faculty supervise students for their clinical skills practice and are supported by well-trained preceptors at those health facilities. A tracer study of employers and graduates of the Aga Khan University found the work-study program to produce skilled and competent bachelor-prepared graduates with high retention rates after graduation [25]. Moreover, several similar studies reported that participation in the work-study program exposes learners to new clinical situations, helps them to integrate clinical knowledge and critical thinking skills, improves prioritization and time management skills, and expands the learner’s communication skills within the interprofessional team [26,27]. Interns cited feeling better equipped to transition into the new graduate role safely and effectively [26]. This research evidence indicates that building an innovative program that offers contextually relevant knowledge, attitude, and skills is more likely to significantly contribute to the quality improvement of healthcare workers by bridging the education-to-practice gap than the traditional model of training.

In Tanzania, nursing and midwifery education is well-regulated and structured into diploma and degree programs; however, challenges persist regarding prolonged training duration, strict entry requirements, and limited curriculum alignment with global competency standards. This review was therefore motivated by the need to analyze Tanzania’s nursing and midwifery educational pathways in relation to international trends to identify gaps and explore opportunities to harmonize local programs with global standards. First, we compared the data on nursing/Midwifery university education in Tanzania with the globe with a focus on entry criteria, duration of the program, teaching procedure, and teaching guiding ideology. Second, the analysis of the dynamics of learning models across generations was conducted. Third, we compared the two teaching and learning models (the work-study model and the traditional model) that are used in Tanzania on their benefits and limitations. Fourth, the pathway of nursing and midwifery education in Tanzania was explored. The inclusion of global comparisons allows benchmarking of Tanzania’s progress against international standards, while the discussion of generational differences and models of training provides insight into how evolving educational approaches, learning frameworks, and career motivations shape the future of the nursing and midwifery workforce.

## 2. Methods

We adopted the hybrid systematic narrative review methodology for literature review to synthesize the study methods and results related to pathways to nursing and midwifery professions in Tanzania, as compared to the globe [28]. A systematic search across multiple databases identified relevant literature, while the narrative synthesis allowed in-depth thematic analysis, contextual interpretation, and identification of patterns, gaps, and trends. The following questions were used to guide this non-systematic narrative literature review: What are the pathways to nursing and midwifery professions in Tanzania as compared to the global context? To answer this question, we reviewed the literature to identify the most significant studies and theoretical foundations about pathways to nursing and midwifery education.

### 2.1. Data Source

A narrative literature review was conducted using multiple databases, including MEDLINE, PubMed, Google Scholar, EBSCOhost, PsycINFO, and Clinical Nursing Databases, to identify both peer-reviewed publications and gray literature published between January 2014 and December 2024. The search strategy employed a combination of keywords and Boolean operators, including “nursing and midwifery”, “admission,” “pathway to the nursing profession,” “delivery mode,” “generative dynamics,” and “learning models.” These terms were used individually and in combination, applying truncation to capture relevant variations. The snowballing technique was further applied to identify additional articles from the reference lists of selected papers. Gray literature sources, including institutional and policy documents, were retrieved from reputable online repositories and organizational websites known for providing reliable information on nursing and midwifery admission criteria, program duration, teaching modalities, and career pathways. The review included studies conducted in East Africa for regional representation, Europe, North America, Australia, and Asia for a diverse global benchmarking and advancement in nursing and midwifery education models. The search was restricted to publications in English, and studies were included based on their relevance to nursing and midwifery educational pathways.

### 2.2. Inclusion and Exclusion Criteria

This narrative literature review involved qualitative studies, quantitative studies, and mixed methods approaches.

The inclusion criteria were (1) published papers or reviews addressing the review topics; (2) studies published in the English language; (3) gray literature on the reviewed topics; (4) studies originating from Tanzania, East Africa, Europe, Asia, Australia, and North America. The reason for choosing these countries was to compare a comparison of literature within the East Africa region with comparing other countries around the globe. The selected countries served as a source for comparison of nursing and midwifery education in Tanzania with the globe. Articles were excluded if the methodology was not clear or duplicated. The details are provided in Figure 1.

### 2.3. Screening

We used the EndNote X9 (Clarivate Analytics, Philadelphia, PA, USA) software to manage search strategies. Two authors (TM and ML) independently reviewed all the titles and Abstracts and extracted the data. Then, the two authors conducted a meeting to reach a consensus on the studies to be included. In cases of disagreements or discrepancies, another author (ES) was contacted to reach a mutual consensus. Screening of the articles began with an examination of methodology, focusing on pathways to nursing and/or midwifery professions, including entry criteria, duration, learning mode, and generation.

### 2.4. Quality Checking

The quality of the included studies was evaluated using a structured checklist adapted from the Critical Appraisal Skills Programme (CASP) for both qualitative and quantitative studies. Each study was independently reviewed by T.M. and M.L., and discrepancies were resolved in consultation with E.S. The appraisal focused on study validity, methodological rigor, data collection and analysis processes, and clarity of findings. Studies were rated as high, moderate, or low quality based on how well they met the CASP criteria. This quality assessment enhanced the credibility and reliability of the review findings.

### 2.5. Data Synthesis

An extraction sheet was developed to extract information from articles, reviews, editorials, and gray literature. Study data were synthesized through a narrative synthesis. Analysis of the articles began with an examination of methodology, keywords, construct, and variables measured. As indicated in Table 1, the data extraction sheet was used to extract the key relevant data, including study purpose, design, sample, methods, and findings reported by study authors and summarized in table form. We synthesized all articles that met the inclusion criteria by comparing and interpreting findings to identify themes and grouping them by similarity.

## 3. Results

Theme 1: Pathway to the nursing profession in Tanzania.

As illustrated in Figure 2, the duration from primary education to diploma takes about 14 educational years. With preparatory years for primary education (kindergarten), it takes about 16 years to achieve a diploma in nursing. The pathway for those who choose direct entry into the nursing profession takes 13 years, while the post-RN BScN program is designed for Registered Nurses and Midwives with a Diploma. Nursing education in Tanzania is regulated by the Tanzania Nursing and Midwifery Council (TNMC), the Tanzania Commission for Universities (TCU), and the National Council for Technical and Vocational Education and Training (NACTVET). After completion of an accredited nursing program, graduates must pass the National Council Licensure Examination to become a licensed registered nurse (RN) or midwife with the TNMC. While NACTVET regulates technical and vocational education, TCU regulates higher education programs offered locally or from foreign countries. As illustrated in Figure 1, there is no career progression on the technical education pathway to the bachelor’s level for health programs. Thus, the only option for those with a Diploma is to join the university education framework, where they encounter stringent TNMC, TCU, and specific university admission criteria.

Theme 2: Comparing Nursing/Midwifery university education in Tanzania with the globe.

In a comparison of Tanzania with other countries entry criteria for nursing and midwifery professions are relatively equal [7,29,30,31]. However, the duration and criteria for admission from diploma to bachelor’s level are disproportionately higher compared to other countries around the world. In Lower-Income Countries (LICs), including Tanzania, the conventional method of teaching by using lecture methods remains dominant, while other countries have moved to blended learning. The details are provided in Table 1.

Theme 3: Dynamics of learning across generations.

Generations are typically defined by the unique cultural, social, and technological experiences that shape the lives of people born during a particular time [32,33,34]. People are influenced by a combination of their generational experiences, personal characteristics, and unique circumstances. An understanding of the generational dynamics can help tailor educational approaches to better meet the diverse needs of learners from different generations. Table 2 describes characteristics and learning models across generations.

Theme 4: Course Delivery Method.

Aga Khan University School of Nursing and Midwifery is offering a five-semester (2.5-year) nursing program for working registered nurses who upgrade to a Bachelor of Science through a work-study model [24]. This program is a flexible program complemented by teaching and learning technology, Blended learning, flipped classrooms, diverse face-to-face teaching, mentorship, use of clinical champions as role models, Moodle platform (online access to teaching and assignment submission), and a digital library that is available for 24 h. As described in Table 3, there are several advantages attached to this program as compared to the traditional training model.

## 4. Discussion

The current narrative review aimed to examine the pathways of nursing and midwifery education in Tanzania in comparison with global standards. The findings revealed that, although Tanzania’s educational pathways share some similarities with international practices in terms of regulation, competency-based training, and professional accreditation, they are considerably different in duration and entry requirements for advancement. The extended training period of four years and mandatory internship requirements contradict the global trends that emphasize that training pathways for nursing and midwifery should be flexible, adaptable, and inclusive [35]. These lead to misuse of resources and a lack of career progression for those who initially chose to join the nursing and midwifery profession.

In Tanzania, the nursing profession is structured around two primary educational pathways leading to professional registration: the diploma program, mainly offered in colleges and regulated by the NACTVET, and the BScN program is offered in universities under the supervision of the TCU. Entry into the bachelor’s program in midwifery is exclusively reserved for diploma holders and is further constrained by the requirements of at least two years of professional experience and strong academic performance in science subjects. Upon completion of either pathway, graduates must undertake the TNMC licensing examination, which evaluates essential professional and technical competencies for practice. However, the stringent entry requirements and limited avenues for career progression present challenges in attracting new entrants and retaining qualified nurses and midwives within the profession.

The entry point to nursing and midwifery programs has been a subject of much debate. In this review, it has been found that in Tanzania, the admission criteria and duration of the program for upgrading nurses and midwives to BScN are beyond the minimum global nursing and midwifery educational requirements [35]. For example, the admission criteria for the year 2023 were “*Three principal passes in Chemistry, Biology and either Physics or Mathematics or Nutrition with a minimum of 6 points such that an applicant must have at least “C” grade in Chemistry and at least “D” grade in Biology and “E” grade in Physics or Advanced Mathematics or Nutrition”* for direct entry to the nursing profession [17]. These entry criteria are higher than the Doctor of Medicine, pharmacy, and DDS program that requires “*three principal passes in Physics, Chemistry, and Biology with a minimum entry of 6 points; such that an applicant must have at least a “D” grade in Chemistry, Biology, and Physics”* [17]. The admission criteria for upgrading nursing and midwifery students were “*Diploma in Nursing with an average of ‘B’’ or a minimum GPA of 3.0. In addition, an applicant must have a minimum of a “D” grade in the following subjects: Mathematics, Chemistry, Biology, Physics, and English O-Level.* However, in the previous year 2022/2023, admissions for upgrading were “*Diploma in Nursing with an average of “B’’ or a minimum GPA of 3.0. In addition, an applicant must have a minimum of a “D” grade in any five (5) non-religious subjects at O-Level*” [17].

Though it is well known that the universities are autonomous in designing, enrollment, and delivery of the programs, the criteria for entry into health programs, especially for nursing and midwifery, come from regulatory bodies and sometimes are contrary to the university curricula. Tanzania’s higher education system had a low gross enrolment rate of the age cohorts of the youth who constitute a pool of potential candidates for university education, with the rate standing at 4.2% which was much lower than the sub-Saharan African average of about 8% [17]. This might have been attributed to the stringent admission criteria that are changing every year. The stringent and frequent change in criteria hinders the recruitment of nurses and limits hopes and opportunities for nurses to pursue their career path to the bachelor’s level and beyond. Lack of opportunity for career development in the nursing professional pathways led to previously registered nurses exiting the profession, job dissatisfaction, burnout, depression, and malpractice. The academic pathways for nurses should be flexible, adaptable, and inclusive to facilitate their career progression [36]. Multicounty experience has indicated that career progression is a strong incentive for nurses and midwives to remain in the workforce [37]. Therefore, there is a need to create a flexible career pathway to upgrade nurses and midwives and set reasonable entry criteria for new entrants to attract and retain them in the profession.

The development of nursing and midwifery education in Tanzania has evolved through a series of policy reforms, professionalization efforts, and alignment with national health priorities. Formal nursing education was first introduced in the 1930s under missionary and colonial hospitals, primarily aimed at producing auxiliary nurses to support medical officers [38]. In the late 1990s and early 2010s, Tanzania introduced university-based BScN and Bachelor of Midwifery programs to elevate academic credibility and professional standing. More recently, the TNMC instituted stringent entry criteria, a mandatory four-year degree, and a one-year internship to ensure clinical competence before professional registration [39]. While these reforms have advanced educational qualities and professional recognition, they have also constrained workforce flexibility and delayed the entry of qualified nurses into the labor market.

The Nursing and Midwifery Act of 2010, among others, states that any person shall be entitled to be registered as a nurse or midwife if they have completed a course of training as a nurse for not less than three years or a midwife for a duration of not less than one year. Currently, in Tanzania, both new entrants and upgrading nurses need to study the Bachelor of Science in Nursing for four years with an additional year of mandatory internship. This is contrary to their own Nursing and Midwifery Act of 2010 and to other countries around the globe, where the duration for upgrading nurses ranges from 2.25 years in Australia [40] to three years in all other countries around the globe [5,35], including in the same region in East Africa [16,41]. In the Tanzanian health professional context, an internship is a program during which graduates work in a hospital under supervision and are paid a monthly salary, which serves as a prerequisite for professional registration. Implementation of an internship program for new graduate nurses plays a vital role in preparing them for better professional performance, enhancing clinical skills, increasing independence, strengthening professional commitment, improving the chances of employment after graduation, and serving as a prerequisite for professional registration. However, in Tanzania, even registered nurses must undergo a mandatory one-year internship program after they graduate from a nursing degree program. In some countries, an internship is part of the training curriculum [42]. Moreover, the mandatory internship program for registered nurses has not been reported elsewhere. Thus, the authors of this study suggest that the implementation of an internship program for employed registered nurses could be beneficial to them if implemented in their respective local hospitals. This will facilitate the retention of nursing staff and offer positive cost benefits to the trainee, the workplace, and the government.

Initially, the diploma was expected to be offered by various technical institutions in Tanzania that are regulated by NACTVET; however, the path to upgrade from diploma to bachelor’s through this path was blocked. Thus, in Tanzania, the university remained the sole route for both new entrants and upgrading nurses and midwives. Thus, the career paths for diploma-registered nurses and midwives were more complex and limited than those available for direct entrants, as initially, they need to meet criteria for technical institutions and criteria to cross the path to the university, which often are conflicting and unfavorable to upgrade from diploma to Bachelor of Science in nursing. Moreover, the complexity of admission criteria for diploma nurses to the bachelor level hampers the WHO standards on the future global goal to shift the initial education of professional nurses and midwives towards bachelor’s university-based education [8]. Moreover, experience from the UK indicates that career pathways need to be linked to competencies with knowledge and skills development appropriate to the articulated career levels [5]. However, some countries, like China, have a duration of up to 5 years for pre-registered nurses, which might be related to the fact that their training follows the medical model [30,43]. Since nurses’ sense of job satisfaction and achievement is tied to career development, education, training, and professional autonomy [2,44], therefore, addressing the pre-requisite qualifications and developing a suitable career pathway and educational frameworks is likely to attract and retain nurses and midwives in the sector.

In Tanzania, as it is with many LICs, the main education delivery for university education is through a traditional model, which is performed primarily through classroom lectures, textbooks, and assignments [45]. Typically, in this model, the theory and practice take place at different times [45,46]. The Aga Khan University in East Africa, specifically, Uganda, Kenya, and Tanzania, for over twenty years, has been running a Bachelor of Science in Nursing program by using the work-study model [24]. Compared to the traditional model, the researcher argues that the work-study program provides an opportunity for industry-institution collaboration, and it benefits all three parties, namely the student, the educational institution, and the industry, in a ‘win-win’ situation [26]. Furthermore, work-study programs provide students with exposure to new clinical situations, integrating clinical knowledge and critical thinking skills, prioritization, communication, and time management skills. The work-study model has been used in several other countries [24,26,47]. For example, Australia for many decades, has used the work-study model in tertiary-level institutions for the preparation of registered nurses and midwives [48]. Studies from the African context found the work-study program to impact positively on nursing and midwifery knowledge, skills, and competencies [24,49]. Moreover, in East Africa, a work-study program has reported a graduate nurse retention rate of nearly 98% [24,47], highlighting the effectiveness of such programs in promoting workforce retention. Furthermore, the work-study program provides exposure to various job demands, job resources, and academic outcomes. Therefore, the Work-Study Program provides a meaningful integration of theory and clinical practice with the potential to improve the quality of education and required competencies.

In this narrative review, we found each generation to possess unique cultural, social, and technological experiences that shape their learning strategies. In Tanzania, the majority of faculty and regulators for nursing education are from Generation X (1960–1980) and Generation Y (1980–1990), who prefer structured learning environments, such as traditional classrooms and lectures, and are comfortable with textbooks and well-organized curricula [33,50]. Unlike the faculty and regulatory generations, Generation Z forms the majority of students entering nursing and midwifery educational programs and subsequently joining the nursing or midwifery workforce [34]. It is believed to be born into a world with constant access to the internet and information, making them digital natives and technology consumers [32]. Moreover, Gen Z learners are passionate about learning through videos, pictures, emojis, and communications through social media [32] but not books, and have a shorter attention span of 8 s as compared to 12 s for Millennials [32,34,50]. Furthermore, Generation Z appreciates active, interpersonal, and hands-on learning opportunities in the classroom as well as in clinical settings [50], and they expect their faculty to engage in the same way [32]. This poses a challenge to faculty and regulators as to whether the current nursing and midwifery curricula, teaching pedagogy, duration of the programs, and clinical teaching settings suit Generation Z’s characteristics and learning preferences. Early research shows that Generation Z is cost-conscious, desires to learn practical skills, is entrepreneurial, and has a more global perspective, combined with advancing technology that makes education more accessible from international and online schools [33,34]. A recent study indicated that Generation Z students preferred graduating in 2 years compared to Generation Y students, who preferred graduating in 3 years or longer. The reason for Generation Z to prefer less duration of study might be related to that many of the jobs, careers, and some of the nursing procedures will change, evolve, or become obsolete in the future due to technological advancement and future generation preferences [33,51]. Some large tech companies have removed four-year degrees from their job requirements, opening the door for employees with self-taught tech expertise but no diploma. This entails that lifelong learning will be a necessity for this and future generations [33,50]. Thus, the current review provides an understanding of the characteristics and learning preferences of Generation Z students and can help faculty, mentors, clinical instructors, and regulators in setting appropriate academic programs and learning environments.

## 5. Limitations

Despite the current review providing valuable information on the pathways to the nursing and midwifery profession and reflection on resource-limited countries like Tanzania, there are some limitations that are worth mentioning. First, the literature search can be considered narrow, leading to few reviewed articles. However, the literature searches were conducted in several databases to increase the reliability of the gathered information. Second, the information in this review might be outdated as nursing education is dynamic. The current study was carried out between January 2014 to December 2024, and changes might have happened after this review period. Third, the current literature is a subjective narrative review that does not follow a strict and systematic methodology for searching, selecting, and evaluating studies. This could have resulted in the absence of some relevant studies or the inclusion of biased studies. However, the credibility and reliability of this literature review were ensured by having clearly defined criteria for the selection of studies and transparent reporting of the findings. Fourth, only studies published in English were included in this narrative review, therefore there might be the omission of some relevant studies that were published in other languages.

## 6. Conclusions

This review demonstrates that there is a need for national nursing and midwifery training standards that are responsive to professional competencies, clients’ needs, and learners’ models. Whilst nursing and midwifery education is regulated in Tanzania, the current admission criteria and duration of the program do not reflect global standards. Moreover, though the national career frameworks are necessary for the recruitment and retention of nurses who are already working in the profession and the attraction of new professionals, the current admission criteria and duration of the nursing and midwifery program in Tanzania are inconsistent and lack quality assurance of the outcomes. Future studies that provide a comparison of curricula and related outcomes between universities in Tanzania would provide a deep understanding of the competencies, teaching models, learning environment, duration, and desired learning outcomes for nursing and midwifery education.

## 7. Recommendations

To attract recruits and increase the education level of registered nurses and midwives, the current study recommends the stakeholders’ involvement in the formulation of admission criteria that relate to nursing and midwifery professional roles and responsibilities. Regulatory bodies for nursing and midwifery university education need to develop educational frameworks that consider the admission and duration for upgrading nurses and midwives from a diploma to a Bachelor of Science. Moreover, there is a need for nursing and midwifery training institutions to consider the generational dynamics and preferred learning models in their curriculum development and implementation.

## Figures and Tables

**Figure 1 nursrep-15-00452-f001:**
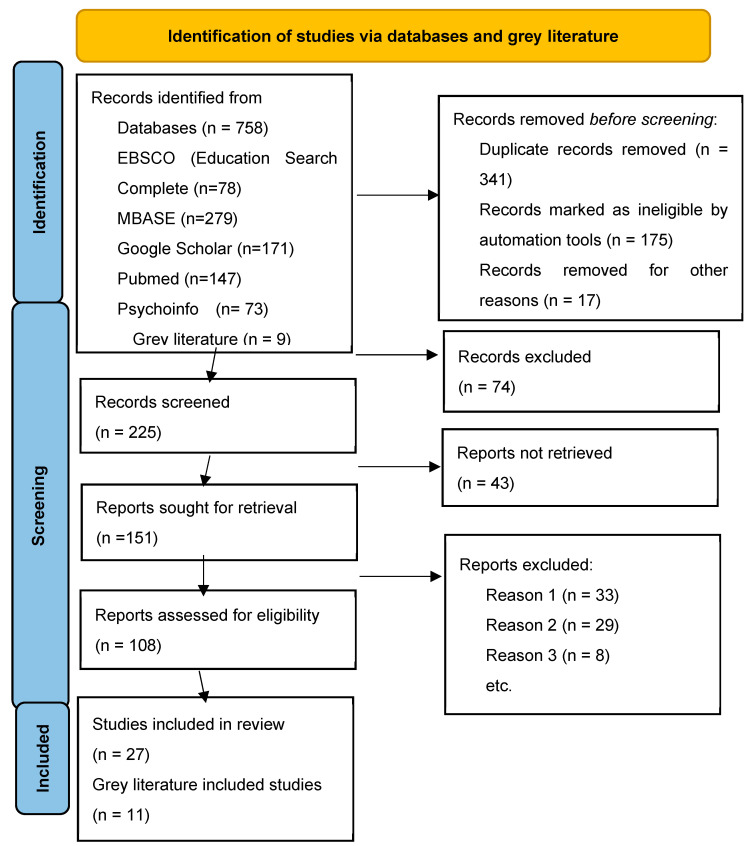
PRISMA 2020 flow diagram illustrating the study selection process, including the number of records identified, screened, excluded, and included in the final qualitative and quantitative synthesis.

**Figure 2 nursrep-15-00452-f002:**
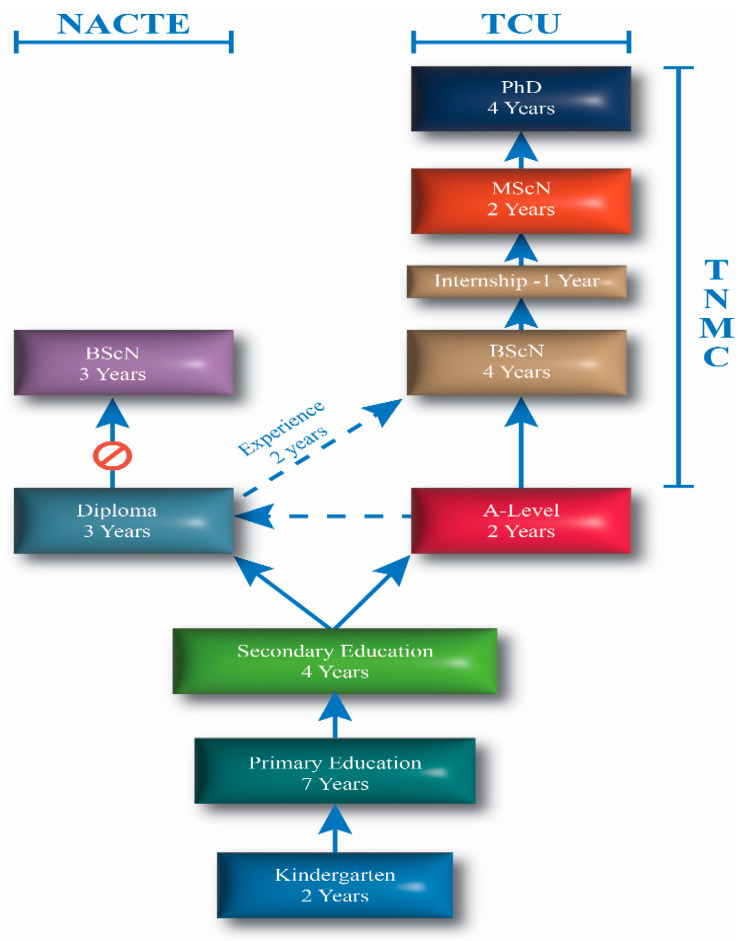
Pathways to the nursing profession in Tanzania, showing the two main regulatory routes governed by the Tanzania Commission for Universities (TCU) for degree programs and the National Council for Technical and Vocational Education and Training (NACTE) for diploma and certificate programs.

**Table 1 nursrep-15-00452-t001:** Comparison of Nursing/Midwifery university education in Tanzania with the globe.

Region	Country	Entry Criteria	Program Duration	Teaching Procedure	Guiding Ideology	Main Teaching Form
East Africa	Tanzania	Direct Entry: Three principal passes in Chemistry (“C”), Biology, and Physics/Mathematics/Nutrition. Upgrading (UPG): Diploma in Nursing with a “B” grade or GPA ≥ 3.0 and a minimum of “D” in five O-Level subjects.	Direct: 4 years UPG: 4 years	Theory → Clinical Practice	Patient-centered care	Lecture
	Kenya	Direct Entry: KCSE Mean Grade C+. UPG: Diploma in Nursing, KCSE Mean Grade C (plain) or Division 3, valid practice license, ≥2 years work experience.	Direct: 4 years UPG: 2.5 years	Theory → Clinical Practice	Patient-centered care	Lecture
	Uganda	Direct Entry: UCE with ≥5 passes or ACEE (Principal passes in Biology, Chemistry, and one subsidiary). UPG: Diploma in Nursing or related field with ≥1 credit.	Direct: 4 years UPG: 3 years	Theory → Clinical Practice	Patient-centered care	Lecture
Europe	United Kingdom	Direct Entry: 10 years of general education. UPG: Diploma (Accelerated BSN, BSN, RN-BSN, or Dual BSN-MSN).	Direct: 3 years UPG: 2–4 years	Theory → Practice → Professional Theory → Clinical Practice	Patient-centered holistic care	Multimedia
America	United States	Direct Entry: High school certificate with Physics, Chemistry, and Biology (≥70%). UPG: Associate or Diploma in Nursing.	Direct: 4 years UPG: 3 years	Child → Adult → Geriatric sequence	Patient-centered holistic care	Multimedia and Self-learning
Australia	Australia	Direct Entry: Australian Year 12 qualification (or equivalent) including English and a Science subject. UPG: Diploma in Nursing.	Direct: 3 years UPG: 2.25 years	Theory → Clinical Practice	Course-oriented	Multimedia and Lecture
Asia	China	Direct Entry: High school completion. UPG: Diploma or Advanced Diploma in Nursing.	Direct: 5 years UPG: 3 years	Theory → Practice → Clinical	Holistic care	Multimedia and Lecture
	Japan	Direct Entry: 12 years of basic education (Mathematics, Science, Japanese & Foreign Languages, Geography, History, Civics). UPG: 4-year Nursing College.	Direct: 4 years UPG: 4 years	Theory → Practice	Holistic care	Self-learning

UPG: Upgrading, UCE: Uganda Council of Education; KCSE: Kenya Council of Secondary Education; ACSEE: Advanced Certificate of Secondary Education: GPA: Grade Average Point; BSN: Bachelor Science in Nursing; MSN: Masters of Science in Nursing; DE: Direct entry.

**Table 2 nursrep-15-00452-t002:** Characteristics and Learning models across the generations.

Generation	Born Between	Characteristics	Learning Models
Silent Generation	1928–1945.	They are known for their hard work ethic and tend to value traditional values and norms.	Traditionalists often prefer structured and formal learning environments. May be more accustomed to teacher-centered education. Value discipline, hard work, and respect for authority.
Baby Boomers	1946–1964.	They are often associated with strong work ethics and loyalty to their employers.	Tend to appreciate in-person, classroom-based learning. May prefer lectures and textbooks as primary learning resources. Value teamwork and collaboration in educational settings
Generation X	mid-1960s–early 1980s.	They are known for their independence, skepticism, and adaptability.	More likely to embrace self-directed learning and independent study. Comfortable with technology but not as reliant on it as younger generations. Value practicality and real-world application of knowledge.
Millennials (Generation Y)	early 1980s–mid-1990s.	They are often characterized as tech-savvy, socially conscious, and interested in work–life balance	Pioneered the use of the Internet and digital technology in education. Tend to value interactive, multimedia-rich, and collaborative learning experiences. They appreciate feedback and personalized learning paths.
Generation Z	mid-1990s–early 2010s.	They tend to be tech-native, diverse, and socially conscious. They also value authenticity and are known for their entrepreneurial spirit.	Grew up in a digital age and are highly tech-savvy. Prefer online and mobile learning, often utilizing a variety of digital tools and resources. Value experiential learning and hands-on experiences.
Generation Alpha	2010–Onwards	They are true digital natives. Value inclusivity, environmental awareness, and global connectedness.	Generation Alpha is expected to be the most technologically connected and educated generation.

**Table 3 nursrep-15-00452-t003:** Comparison between the work-study model and with traditional training model.

	Key Features	Advantages	Limitations
Work-study model	Students study while continuing to work. The model is Student-Centric It combines theory with real-world practical experience.	Enables students to maintain their full-time employment and the associated income. Enable students to work within their proximity without impacting adversely on the work in their respective working place. Immediate practical skills development Nursing industry-relevant experience Exposure to real-world problem-solving skills. Utilize research to ensure a sound evidence-based practice	Student learning needs may not be congruent with his/her working facility. Lack of routine supervision in individual practice Increased demand to balance work and study
Traditional Model	Education is delivered primarily through classroom lectures, textbooks, and assignments. Theory and Practice takes time at different times	Strong foundation in theoretical knowledge. Effective for subjects that require deep conceptual understanding.	Limited practical application. Potential for disengagement due to theoretical overload. Limited exposure to real-world problem-based learning

## Data Availability

All data generated or analyzed during this study are included in this published article. No new data were created or analyzed in this study, as it is based entirely on previously published literature.

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
