# Peer review of "Pathways to Nursing and Midwifery Education in Tanzania with Reflection to the Global Perspectives: A Narrative Review"

_nursrep, 2025, doi:10.3390/nursrep15120452_

Round 1

Reviewer 1 Report

Comments and Suggestions for Authors

Manuscript Title: Pathway to Nursing and Midwifery Education in Tanzania with reflection to global perspective: A narrative review

Manuscript ID: nursrep-3908327

1-    In the abstract, the authors mention reviewing papers until 2023; however, the number of papers reviewed is missing from the manuscript. Also, Please mention the complete study duration.

2-    The introduction sounds good; however, minor modifications are required in terms of the motivation behind this study and the most updated/recent studies in the introduction section.

3-    The abstract mixes methods and results poorly and is not structured properly. Please revise and correct it.

  1. Please correct the spelling of “Background” in the abstract.

5-    Mid- income countries or low income countries, kindly revise with “low-income countries”

6-    A critical step in any review (systematic, integrative, or narrative) is missing. Without evaluating the quality of the included studies, the conclusions lack reliability.

7-    Many references are too old; kindly update and replace them with updated studies.

8-    Strong claims such as “Tanzania had the longest duration globally” are made without a robust comparative dataset. The evidence tables were incomplete and not standardized.

9- The conclusion and the abstract section sound good.

10- Kindly revise lines 271–311 to enhance clarity and readability.

Author Response

We sincerely thank the Editor and all anonymous reviewers for their time, effort, and valuable comments, which have significantly improved the clarity, structure, and overall quality of our manuscript. We have carefully considered and addressed all comments as detailed below, with corresponding revisions made in the indicated with RED colour ad line numbers in sections of the manuscript.

Reviewer 2 Report

Comments and Suggestions for Authors

Though this review article addresses an important topic, it is not yet ready for publication. In its current form, it exhibits shortcomings in both content (depth of methodology and discussion, clarity of focus) and structure (language and presentation). The authors need to undertake a comprehensive revision focusing on the points detailed below. In particular, attention should be paid to improving the clarity of the methodology section, sharpening the focus of the literature synthesis, presenting the findings in a more cohesive manner, and correcting the language.

I believe that implementing changes in line with the criticisms raised in my referee report will significantly enhance the quality of the article. The authors may resubmit the work to the journal for re-evaluation after a major revision. In this process, if content consistency is ensured and the level of academic language is elevated, I am of the opinion that the article can reach a level that contributes meaningfully to the literature in the field.

Below, in light of the above evaluation, the main issues observed and suggestions for improvement are summarized in numbered points:

  1. In the introduction, the aims and scope of the article should be stated more clearly. It should also be explained to the reader why subtopics such as global comparisons and generational differences are addressed. This clarification will strengthen the focus of the article.
  2. The English phrasing error in the title of the article (using “with reflection on” instead of “with reflection to”) should be corrected, and the typos in the abstract should be fixed (for example, use "Background" instead of "Backround"). Ensuring the title and abstract are error-free will improve the article’s first impression and clarity.
  3. The literature search strategy needs to be presented in more detail and with transparency. Clearly state which databases were searched, with which combinations of keywords, and covering which years. Any search limitations (language, region, year, etc.) should be explicitly mentioned. This level of detail will allow readers to understand the scope of the literature review and trust its thoroughness.
  4. It should be explained by what criteria the comparison countries outside Tanzania (e.g., Kenya, Uganda, the UK, the USA, China, etc.) were selected. For example, criteria such as representativeness or data availability could be given to show that the choice of countries is grounded in rationale and not arbitrary. This will strengthen the validity of the cross-country comparisons.
  5. The cross-country comparison table (Table 1) should be made more organized and easier to follow. Each column and row should be formatted in a way that allows the reader to make comparisons at a glance. If necessary, consider splitting the table into two smaller tables or summarizing important comparisons in the text. Ensure that all data presented in the table are up-to-date and drawn from accurate, reliable sources.
  6. If the findings regarding the work-study model (e.g., a 98% graduate retention rate) are based on results from literature, they should be explicitly stated and supported by references. If certain points are the authors’ own interpretations or suggestions, these should be clearly marked as such—preferably placed in the discussion or conclusion section rather than the results section. This way, the reader can easily distinguish between objective findings from the literature and the authors’ personal recommendations.
  7. In the discussion section, while criticizing current practices in Tanzania, also briefly address the possible rationale or historical context behind those practices. For example, you might note that although stringent entry criteria were intended to improve quality, they may have had the opposite effect. This approach adds an objective balance to the discussion and can strengthen the persuasiveness of your policy recommendations by acknowledging why certain policies existed in the first place.
  8. The recommendations given in the final section (such as stakeholder participation, regulatory boards establishing a framework, and considering generational dynamics in the curriculum) are pertinent. However, instead of presenting these suggestions as a bullet-point list, consider integrating them smoothly into the discussion or prioritizing them by importance. It should also be made clear which findings each recommendation is based on. For example, a concrete recommendation could be: “Shortening the transition from diploma to bachelor’s degree (reducing it to 3 years) will both be in line with the spirit of the national law and increase professional motivation.” Including such specific, evidence-backed recommendation statements will make the conclusion more impactful.
  9. The entire article should undergo a thorough language and grammar check. Identified errors need correction: fix any subject-verb agreement issues, ensure consistent use of verb tenses, and correct word choice errors (e.g., using “high” vs. “long” appropriately when referring to quantities or durations). Also, eliminate preposition errors (such as confusion between to/on, in/at, etc.). If needed, it is advisable to use a professional English editing service to polish the language.

If the revisions outlined in the above points are carried out, the article’s scientific contribution level and writing quality will be significantly improved, bringing it to a publishable standard. I strongly recommend that the authors give due attention to addressing these points during the revision process.

Author Response

(The authors gave the same response as above.)

Reviewer 3 Report

Comments and Suggestions for Authors

Author Response

(The authors gave the same response as above.)

Reviewer 4 Report

Comments and Suggestions for Authors
  •  Some of the keywords used for the literature search were too general, especially when they were applied individually rather than in combination. For example, the keyword “admission” is too broad; a more specific term such as “admission criteria nursing” or a similar phrase would have been more appropriate. Others were too unspecific, for example the keyword “delivery mode.”

  • The significance of the study is clearly articulated by the author, noting that nursing and midwifery professionals constitute approximately 80% of the global health workforce and play a vital role in the health system.

  • In the literature review, only qualitative statements are made (e.g., graduates of work-study programs are described as “competent” and having high retention rates). The argumentation would be stronger if concrete figures were provided to quantify how the training systems impact the health system—for example, in terms of the number of graduates, improvements in patient care, reduction of errors, or meeting workforce demands.

  • Comparisons with the global context are only truly meaningful if it is known that the systems in the reference country are successful—expressed quantitatively.

  • It would be scientifically appropriate not to focus solely on the negative aspects of the extended training pathway in Tanzania, but also to highlight potential advantages, such as providing students with more time to acquire theoretical knowledge and practical skills, which could enhance their overall competence and preparedness for professional responsibilities.

Author Response

(The authors gave the same response as above.)

Reviewer 5 Report

Comments and Suggestions for Authors

The article submitted for review addresses a significant organizational and educational issue in the healthcare system: reflecting on the path to nursing and midwifery education in Tanzania from a global perspective. Nurses and midwives play a significant role in the healthcare system worldwide. These professions are constantly evolving toward changing models of healthcare delivery, emphasizing evidence-based practice. This article is the result of a narrative literature review.

Abstract: Divided into sections (Background/Objectives; Methods; Results; Conclusions), which briefly introduce the article's topic and provide an overview of the research reviewed. The number of keywords reflects all the issues discussed in the article.

In the introduction, the authors noted that the Sustainable Development Goals (SDG 3) address the development of healthcare workers through appropriate training, education, and skills tailored to the needs of the population. In some high-income countries, a bachelor's degree has been established as the minimum entry requirement for nursing and midwifery, including Poland, although the authors did not analyze nursing and midwifery education in all European countries. They then discuss the very stringent entry criteria for nursing and midwifery in Tanzania, which hinder new entrants to the profession and limit professional development. The authors emphasize that recent years have seen significant changes in healthcare information technology and the demands on healthcare systems, requiring innovative models of nursing and midwifery education to address these changes. They point to research evidence suggesting that creating an innovative program offering relevant knowledge, attitudes, and skills is more likely to improve the quality of healthcare professionals.

The aim of this narrative review undertaken by the Authors was to analyze the global nursing and midwifery education pathway, with a focus on the Tanzanian context. Data on nursing and midwifery education at universities in Tanzania were compared with data from other countries worldwide, with particular emphasis on admission criteria, program duration, and teaching procedures. The dynamics of learning models across generations (work-study and traditional models) used in Tanzania were analyzed, examining their benefits and limitations. Furthermore, the educational pathways for nurses and midwives in Tanzania were examined.

Research Method: The Authors conducted a narrative literature review based on databases containing published nursing research and gray literature through 2023. Several databases were searched, and search rules (keywords) and inclusion and exclusion criteria for studies were established, as presented in Figure 1. Endnote software was used to manage the search strategies. Article selection rules and a data extraction spreadsheet for extracting information from the articles were presented correctly.

Results: The Authors grouped these into four themes and summarized them in detail, including a graph, tables, and narrative. This review revealed that although the educational pathways for nurses and midwives in Tanzania were similar to many other countries worldwide (the authors compared nine countries), the admission criteria and program duration, particularly for nurse and midwife advancement, were very high and inappropriate for these professions. Tanzania had the longest duration of education (four years), with an additional year of mandatory internship.

Discussion: The Authors primarily noted that advanced nursing and midwifery programs in Tanzania lead to inappropriate use of resources and a lack of career development opportunities for those who choose to pursue nursing and midwifery. Furthermore, the Tanzanian higher education system is characterized by low university enrollment rates. At the same time, research indicates that nursing and midwifery education pathways should be flexible, adaptable, and inclusive to facilitate professional development, as demonstrated by analysis of education systems in other countries. This review also provides insight into the characteristics and educational preferences of Generation Z, Y, and X students.

In conclusion, the Authors concluded that there is a need for national standards for nursing and midwifery education that address professional competencies, patient needs, and student learning models. They demonstrated that although nursing and midwifery education in Tanzania is regulated, the current admission criteria and program duration do not reflect global standards.

The article uses 50 items of current scientific literature - all of them are cited.

The authors also considered the limitations of their own research and the recommendations resulting from the undertaken studies, which is of great value.

In my opinion, all sections of this article were prepared comprehensively, clearly, and in a manner that raises no substantive or ethical concerns. I highly recommend this article for publication. In my opinion, the article requires no additions. The practical aspect of the study is worth emphasizing, as the results of the authors' review provide key information for the regulation and implementation of nursing and midwifery education both in Tanzania and abroad. They point to the need to create a flexible career path that will enable nurses and midwives to advance professionally and to establish reasonable entry criteria for new entrants to attract and retain them in the profession.

I highly recommend this article for publication.

Author Response

(The authors gave the same response as above.)

Round 2

Reviewer 1 Report

Comments and Suggestions for Authors

Accept 

Reviewer 3 Report

Comments and Suggestions for Authors

Thank you for addressing each suggested revision. The manuscript is very comprehensive.